# Chemical Recycling of Vacuum-Infused Thermoplastic Acrylate-Based Composites Reinforced by Basalt Fabrics

**DOI:** 10.3390/polym14061083

**Published:** 2022-03-08

**Authors:** Inès Meyer zu Reckendorf, Amel Sahki, Didier Perrin, Clément Lacoste, Anne Bergeret, Avigaël Ohayon, Karynn Morand

**Affiliations:** 1Polymers Composites and Hybrids (PCH), IMT Mines Alès, 30100 Alès, France; ines.meyer-zu-reckendorf@mines-ales.fr (I.M.z.R.); amel.sahki@mines-ales.fr (A.S.); clement.lacoste@mines-ales.fr (C.L.); anne.bergeret@mines-ales.fr (A.B.); 2SEGULA Technologies, 69200 Vénissieux, France; avigael.ohayon@segula.fr (A.O.); karynn.morand@segula.fr (K.M.)

**Keywords:** recycling, thermoplastic composites, basalt fibers, ultrasound, mechanical properties

## Abstract

The objective of this work was to compare the material recovered from different chemical recycling methodologies for thermoplastic acrylate-based composites reinforced by basalt fabrics and manufactured by vacuum infusion. Recycling was done via chemical dissolution with a preselected adapted solvent. The main goal of the study was to recover undamaged basalt fabrics in order to reuse them as reinforcements for “second-generation” composites. Two protocols were compared. The first one is based on an ultrasound technique, the second one on mechanical stirring. Dissolution kinetics as well as residual resin percentages were evaluated. Several parameters such as dissolution duration, dissolution temperature, and solvent/composite ratio were also studied. Recycled fabrics were characterized through SEM observations. Mechanical and thermomechanical properties of second-generation composites were determined and compared to those of virgin composites (called “first-generation” composites). The results show that the dissolution protocol using a mechanical stirring is more adapted to recover undamaged fabrics with no residual resin on their surface. Moreover, corresponding second-generation composites display equivalent mechanical properties than first generation ones.

## 1. Introduction

### 1.1. Thermoplastic Composite Materials in the Automotive Industry

In 2019, the global market of polymer-reinforced composites materials reached 17.7 megatons in volume with a value of $86 billion. In virtually every region of the world and every application sector, the composites market is growing in both volume and value. China (28%) and North America (26%) remain the largest markets in terms of volume, ahead of Europe (21%) and the rest of Asia (19%) [1].

The main industries using composites, in volume, are transportation (28%), ahead of construction (20%), electronics and electrical (16%), and pipes and tanks (15%) [1]. The automotive industry is one of the largest users of composites in the transportation sector owing to demand for weight reduction in vehicles; this is mainly driven by the demand for better fuel efficiency and reduced CO_2_ emissions in order to comply with EU legislation. From 1 January 2020, this legislation set an EU fleet-wide target of 95 gCO_2_/km for the average emissions of new passenger cars and an EU fleet-wide target of 147 gCO_2_/km for the average emissions of new light commercial vehicles registered in the EU [2]. Moreover, a reduction in weight also implies lower environmental impacts (from <130 gCO_2_/km in 2015 to <95 gCO_2_/km by 2021). Composites can offer some lightening compared to other traditionally dominant structural materials (steel, aluminum, etc.) ranging from 15–25% for glass fiber reinforced polymers (GFRP) to nearly 25–40% for carbon fiber reinforced polymers (CFRP). The benefits of lightweight solutions can be translated into potential savings of 8 million tons of CO_2_ per year in the EU wide vehicle fleet [3].

Consequently, since the 2010s, car makers have been using composite materials for passenger cells (CFRP), car roofs (both CFRP and GFRP), and fluid filter module (GFRP) of their cars [3,4]. Currently choosing CFRP as a chassis material for vehicles is a good solution because of its high strength and low weight. As the production cost of this type of material tends to decrease, different types of CFRP are present in a vehicle chassis, such as the header above the windshield, door sills, transmission tunnel, front-to back and left-to-right roof reinforcement tubes and bows, the B-pillar between front and rear doors which provides structural support for the vehicle roof panel, the C-pillar (the vertical structure behind the rear door), and rear parcel shelf [4]. GFRP have been gradually used for impact-absorbent automotive parts (such as instrument panel and inner door modules).

Glass fibers are by far the most used reinforcement (88%) all composite markets combined [1], but other mineral fibers have been emerging over these last years. Among them, basalt fibers are one of the most promising.

Basalt fibers are made of solidified lava. The extrusion of this mineral allows the production of basalt fibers. This type of fiber has been widely used in the production of reinforced concrete for floors, construction, and even roads. Currently, this fiber is attracting increasing interest because of its lower cost than glass fibers for equivalent mechanical properties and its better alkaline resistance than E-glass fibers. In a review on basalt fibers, Fiore et al. [5] reported higher mechanical properties for basalt fibers compared to glass fibers with an elastic modulus of 89 GPa, and a tensile strength of 2.8 GPa (76 GPa and 1.8–2.5 GPa, respectively, for E-glass fibers). As the density of basalt fibers (2800 kg/m^3^) is higher than those of glass fibers (2560 kg/m^3^), equivalent specific mechanical properties are obtained for basalt fibers compared to glass fibers (around 30 GPa per kg/m^3^).

In the field of composite materials, thermoset resins are the most used (61%) ahead of thermoplastic resins (38%). Thermoplastic composites have become a real industrial alternative to traditional thermoset composites due to several benefits—among them, the fact that they are recyclable and take part to circular economy [6]. At present, the more commonly used thermoplastic matrices in the automotive industry are polypropylene (PP), polyamide 6 (PA6), polyamide 6.6 (PA66), polyoxymethylene (polyacetal) (POM), high density polyethylene (HDPE), poly(methyl methacrylate) (PMMA), polycarbonates (PC) and polyvinyl chloride (PVC), acrylonitrile butadiene styrene (ABS), and polyetheretherketone (PEEK) [7,8].

Elium^®^ resin is a new thermoplastic polymer produced by Arkema Co. (Paris Villepinte, France) since 2014. It is obtained from a low viscosity reactive mixture of polymer solutions of methacrylic monomers (methyl methacrylate (MMA), alkyl acrylic) and acrylic copolymer chains (viscosity of 100 cPs) and possibly other comonomers. Its polymerization is activated by a dibenzoyl peroxide as a thermal initiator and can be carried out at room temperature in the presence of an iron salt catalyst [9]. Elium^®^ resin is a resin adapted to the vacuum infusion process at room temperature. Regarding its mechanical properties, the tensile modulus is around 3.3 GPa and tensile strength is around 76 MPa with an elongation of 6% [6].

Elium^®^/basalt fibers composites appear to be a promising composite alternative, combining good specific mechanical properties for transportation applications and recycling potential.

### 1.2. Fiber Reinforced Composites Recycling Techniques

Different types of material recoveries could be employed to value the components of composites.

Firstly, mechanical recycling processes consist of crushing the materials after an initial cutting step into small pieces, a step common to all recycling techniques. This mechanical process does not separate the fiber from the resin. Mechanical shredding has been applied more to GFRC, particularly SMC (Sheet Molding Compound) and BMC (Bulk Molding Compound), but studies on CFRC also exist [10].

In addition, several thermal recycling processes allow the recovery of fibers by pyrolysis, combustion, co-incineration, fluidized bed, and molten salt pyrolysis. The principle is to expose the material to high temperatures under oxidant or inert atmospheric conditions to decompose the matrix.

The third technique of material recovery is chemical recycling, which is often associated to solvolysis, defined by the capability of the solvent to break the macromolecular chains as chemical dissolution of the matrix corresponds to the disentanglement of the chains thanks to solvation. There is therefore no breaking of macromolecular chains unlike solvolysis. This technique offers a large number of possibilities with a wide range of solvents, temperatures, pressures, and catalysts.

Chemical recycling performed on acrylate (Elium^®^) based composites reinforced by basalt fabrics will be evaluated in this work.

### 1.3. Chemical Recycling of Acrylate-Based Composites

The solvents considered able to dissolve PMMA are also able to dissolve Elium^®^ resin because of the similar chemical structure of both acrylic polymers. The properties of the different solvents can be controlled by the operating conditions (temperature, pressure, and reaction time) imposed on the process.

Evchuk et al. [11] studied the possibility of existing correlation between the PMMA dissolution rate and the structure or properties of the solvents, including polarity. Solvents such as trichloromethane, trichloroethylene, 1,4-dioxane, cyclohexanone, acetophenone, ethyl acetate, pentyl acetate, and dimethyl formamide were studied by these authors at a temperature range of 30–70 °C. However, no correlation was highlighted. Trichloroethylene was pointed out by these authors to be the best solvent for PMMA, while trichloromethane was the poorest. For Evchuk et al. [11], polar solvents, such as ethyl acetate, tetrahydrofuran, and cyclohexanone, are adapted to dissolve PMMA.

In the works by Tschentscher et al. [12] and Gebhardt et al. [13], these latter solvents were also used to recycle CFRP based on Elium^®^ 150 resin involving specimens of 200 mm × 180 mm. These latter were placed in a closed container with a selected solvent (acetone, acetophenone, ethyl acetate, or xylene). These composites were tested for two “composite:solvent” ratios (1:10 and 1:20) and for three dissolution times (24 h, 48 h, 72 h) according to the solvent nature. The authors stated that a solvent ratio of 1:20 is recommended for a time-saving dissolution process (24 h). Ethyl acetate represents an environmentally friendly alternative to the established room temperature-based recycling process using acetone. The amount of Elium^®^ recovered as well as the amount of Elium^®^ remaining on the fibers were close for both solvents. Nevertheless, Elium^®^ glass transition was nearer to virgin Elium^®^ one in the case of ethyl acetate than acetone. As concerns acetophenone, it was considered by the authors to be a good solvent at higher temperatures.

The review of Oliveux et al. [10] reported several technologies for composites recycling in order to recover fibers. Among them, Adherent Technologies, Inc. (ATI) (Albuquerque, NM, USA) in USA [14,15] use a wet chemical breakdown to recover carbon fibers from industrial waste composites as well as from end-of-life composite materials. From comparison with different recycling techniques, the authors conclude that a process using a low temperature and low pressure is the most interesting in terms of quality of the recovered fibers and of price. In France, Innoveox [16] proposes a technology based on supercritical hydrolysis with the possibility to control the solvent properties and the reaction rates by conducting pressure manipulations. Supercritical water (temperature > 374 °C and pressure > 221 bar) has been mainly applied to CFRC in order to recover carbon fibers of good quality without paying much attention to the products of the resin degradation. SACMO (Holnon, France) [17] has also been interested in the composite solvolysis process and filed in 2014 a patent proposing a device for solvolysis treatment of a solid composite material to extract fibers from the treated material. The reactor can have a volume from 25 to 400 L. Pressure and heating can be applied. In addition, Panasonic Electric Works Co. (Tokyo, Japan) [18] has shown a willingness to exploit their hydrolysis process to recycle 200 tons of GFRC (based on unsaturated polyester resin) manufacturing waste per year. Wastes are treated at 230 °C with subcritical water at 2.8 MPa with additives (NaOH, KOH) for 4 h.

### 1.4. The Use of Ultrasounds as Processing Aids for Solvolysis

For more efficient technologies and in an environmental friendlier approach, mechanical stirring and heating are unfavorable techniques that need a non-negligible amount of energy to be processed. It is the reason why ultrasound was preferentially used to replace a commonly used stirring mechanism during the recycling protocol as a greener alternative.

Das et al. [19] proposed a sono-chemical method to recover carbon fiber from CFRP. Epoxy/carbon fibers composites are treated with a mixture of dilute nitric acid and hydrogen peroxide in the presence of ultrasound. Specimens of 30 mm × 25 mm × 2 mm are immersed for a pretreatment in the mixture until the samples are fully swollen. The solid-to-fluid ratio is maintained at 1:60. Then the sonication (470 kHz, 65 °C) begins in a water bath. The composite layer separation begins after 4 h. The total duration of the treatment is 8 h. The recovered fibers have little or no epoxy resin on the surface and their tensile strength is comparable to virgin fibers. According to Das et al., the use of ultrasound in aqueous solutions leads to cavitation. Microbubbles implode on the surface of the composite and facilitate the degradation of the resin first on the surface and then in the core of the composite, also by solvent diffusion.

Desai et al. [20] have already studied ultrasounds in order to depolymerize polymers such as polypropylene in p-xylene and decalin solvents. The degradation of PMMA using ultrasounds has also been studied. The authors have investigated the effect of alkyl groups’ (such as methyl, ethyl or butyl groups) substituent on the ultrasonic degradation of PMMA, PEMA, and PBMA at 30 °C in toluene. The degradation rate constantly increases with an increase in the number of carbon atoms in the alkyl group [21].

In terms of degradation mechanism, homolytic and/or heterolytic cleavage of a covalent bond may occur. However, the breakage of a C–C bond in the macromolecule is the most common mechanism. Yan et al. [22] established a degradation mechanism of aqueous solution of carboxylic curdlan by ultrasound treatment. The ultrasound treatment (15 min) implies a shear force that is responsible for the disaggregation of polymer clusters. Non-covalent intra- and intermolecular bonds are broken. As the time of the ultrasound treatment increases (30 min), chains are further split, and shorter random coil chains are created. A degradation process thus occurs.

This literature review revealed that, to the best of our knowledge, no investigations have been carried on chemical recycling of Elium^®^/basalt fabrics composites as well on an academic and industrial point of view. Therefore, this paper provide new results on this challenging new composite material and the best way to recover either the Elium^®^ resin or basalt fabrics to produce new composites that will be called “second generation” composites. To lower the carbon footprint of the process, an environmental-friendly solvent and room temperature dissolution procedure will be chosen. In addition, this paper will analyze the influence of ultrasound on the kinetics dissolution of Elium^®^ resin to provide a greener approach, as compared to mechanical stirring and heating.

## 2. Materials and Methods

### 2.1. Materials

This study focuses on basalt woven sized fibers in plain weave. This fabric supplied by Basaltex (Wevelgem, Belgium) has a basis weight of 220 g/m^2^ with a density of 2.65 g/cm^3^ and a fiber diameter of 13 µm. The resin Elium^®^ 150 was provided by Arkema (Lacq, France). The resin has a low viscosity of 100 mPa.s and a density of 1.01 g/cm^3^. According to the producer, its composition is a mixture of a methyl methacrylate copolymer (70–90 wt%), citral or 3,7-dimethyl-2,6-octadienal (≤5 wt%), hydro-treated light paraffinic distillates (petroleum) (≤2 wt%), and other additives. Perkadox^®^ CH-50X dibenzoyl peroxide, provided by AkzoNobel, was added to the resin to initiate the radical polymerisation (2 wt%). After 30 min at room temperature, an exothermic reaction occurred so that no post-curing is required.

### 2.2. Composites Processing

“First generation” composites of 41 cm × 41 cm were manufactured by resin infusion at room temperature with 10 plies of fabric. Elium^®^ resin and initiator (2 wt%) were mixed, and then infused under vacuum outlet until the surface of the fibres was covered with resin. The infused plate was left to polymerize at room temperature for 24 h without any further post-curing. The composites obtained have an average thickness of about 1.68 mm and a mass fraction of 71% and volume fraction of 48% of fibres. The density was measured through Micrometrics gas pycnometer with helium gas. It was 2.01 ± 0.02 g/cm^3^.

“Second generation” composites were also manufactured by resin infusion at room temperature with 10 plies of fabric. For each sample, all the 10 plies belong to the same recovered sample from the first-generation composites. The composites from recycled fabrics have been infused in the dimensions necessary for the tensile tests, i.e., 125 mm × 25 mm under the same processing parameters used for the first-generation counterparts.

### 2.3. Dissolution Parameters

*Parameter selection.* Different parameters were studied: the choice of solvent, the composite/solvent ratio, the dissolution time, the dissolution temperature, and the presence of agitation or not. In this article are presented the results of the dissolution in acetone, comparing the influence of different composite/solvent ratio, time of dissolution and the use of mechanical agitation, ultrasound, or simply dipped in the solvent and are described in Table 1. For this purpose, the dissolution devices were adapted according to the protocol in order to obtain optimal results.

*Sample preparation*. For dissolution experiments, composites were cut using a circular raw. The dimensions were limited to a few square centimeters (from 6 cm × 10 cm and 12.5 cm × 2.5 cm) for a good fixation into the reactor.

*Dissolution devices*. Figure 1 shows the experimental device used for dissolution experiments using mechanical stirring (60 rpm). A mesh size of 1–2 mm sieve with a folded into a cylinder shape is placed at the bottom of the reactor, around the dipping paddle, in order to avoid contact of the paddle with the composite samples. For dissolution experiments using ultrasounds, two composite samples (13.5 cm × 3.5 cm) were dissolved in an 800 mL beaker with 800 mL acetone, as shown in Figure 2. The power of ultrasounds was fixed at 550 W, and the frequency was around 50/60 Hz (Elmasonic S, Lille, France).

Different experimental conditions summarized in Table 1 were applied during this study in order to compare: (i) the effect of the composite/solvent ratio, (ii) the effect of the use of mechanical agitation versus the use of ultrasounds, and (iii) the time of dissolution. The dissolutions were carried out in acetone SLR, Fisher Chemical. For each dissolution experiment, five samples were used, and each experiment was investigated in triplicate

*Parameter determined after dissolution*. After each dissolution time, the basalt fibers were extracted and dried at 60 °C in an oven in order to evaporate any remaining acetone. The solvent also containing the dissolved resin were placed in a rotary evaporator in order to extract the resin in solid form and then dried at 60 °C in an oven.

The percentage dissolution is defined as the rate of resin recovered from the initial composite before dissolution and noted %*D*. It was calculated after each test, as per Equation (1):(1)$$\%D=\frac{m_{\mathrm{recovered}\;\mathrm{resin}}}{m_{\mathrm{initial}\;\mathrm{resin}}}\times 100$$

A dissolution series corresponds to five samples dissolved during five different dissolution times. Each series is repeated three times. The samples in a series are taken from the same initial composite plate.

### 2.4. Characterization Techniques

#### 2.4.1. Determination of Fiber and Residual Resin Contents

Loss on ignition tests were performed on composites to determine either the reinforcement content of first-generation composites or the residual resin content on recycled fibres after resin dissolution. The tests were carried out according to the NF EN ISO 1172 standard.

About 2 g of samples were placed in a ceramic crucible and calcinated at 600 °C for 30 min. Each sample was tested in triplicate.

The weight percentage of fiber and resin, and volume fibre contents of the composites were calculated according to Equations (2)–(4), respectively:(2)$$\;W_{f}=\frac{m_{f}}{m_{c}}\times 100=\frac{\mathrm{mass}\;\mathrm{composite}\;\mathrm{after}\;\mathrm{test}\;}{\mathrm{mass}\;\mathrm{composite}\;\mathrm{before}\;\mathrm{test}}\times 100$$
(3)$$\;W_{f}=\frac{m_{f}}{m_{c}}\times 100=\frac{\mathrm{mass}\;\mathrm{loss}\;\mathrm{after}\;\mathrm{test}}{\mathrm{mass}\;\mathrm{composite}\;\mathrm{before}\;\mathrm{test}}\times 100$$
(4)$$V_{f}=\frac{W_{f}/\rho_{f}}{W_{f}/\rho_{f}+W_{r}/\rho_{r}}\times 100$$
where *W_f_* (%) is the weight percentage of fibre, *W_r_* (%) is the weight percentage of resin, and *V_f_* (%) the volume percentage of fibre in the composite. *M_r_* (g) and *m_f_* (g) are the masse of resin and fibres, respectively. *ρ_r_* and *ρ_f_* are the density of the resin and fibres of basalt (g/cm^3^).

The percentage of residual resin is defined as the rate of resin present on the fibers resulting from the dissolution, noted %*RR*, and measured by the loss ignition test. It was calculated with Equation (5):(5)$$\;\%RR=\frac{\mathrm{mass}\;\mathrm{of}\;\mathrm{fibers}\;\mathrm{before}\;\mathrm{test}-\mathrm{mass}\;\mathrm{of}\;\mathrm{fibers}\;\mathrm{after}\;\mathrm{test}}{\mathrm{mass}\;\mathrm{of}\;\mathrm{initial}\;\mathrm{composite}}\times 100$$

#### 2.4.2. Physicochemical Characterization of Recovered Resin after Dissolution

##### Fourier Transform InfraRed Spectrometry

FTIR (Fourier Transform InfraRed) analysis was performed using a Vertex 70 FT MIR spectrometer from Bruker with an ATR (Attenuated Total Reflection) unit equipped with a diamond crystal. The resolution was at 4 cm^−1^, 16 scans for background acquisition, and 32 scans for the sample spectrum. Most of the samples were directly analyzed on the crystal. Some were cut to obtain better acquisition, as ATR is sensitive to surface aspects. Analyzed surfaces were cleaned with ethanol and left to dry. Spectra were acquired from 4000 to 400 cm^−1^ and analyzed, thanks to the OPUS software provided with the spectrometer.

##### Gel Permeation Chromatography

(GPC) was performed in order to determine the macromolecular chain weight of Elium^®^ resin after dissolution. The tests were performed on the Varian 390-LC device with refractive index (RI) and viscometry detectors. Two columns were used: PLgel 5 microns and Mixed-D 300 mm × 7.5 mm. The eluent was THF, and the flow rate was 1 mL/min. The temperature of the columns was 30 °C. The samples of virgin and recycled Elium^®^ were ground and then dissolved in THF at a height of 10 mg for 1 mL of solvent and 0.05% toluene. After half a day and under ultrasound, the samples were dissolved. There was one test for each sample.

#### 2.4.3. Fabrics and Composites Observations through Scanning Electron Microscopy and Porosity Measurements

Scanning electron microscopy (SEM) was investigated to observe the fabric surface after dissolution in order to confirm or not the presence of residual resin.

SEM was also carried out on composites of the first and second generation to examine the resin impregnation of the fabrics.

MEB FEI QUANTA 200 FEG was used in both cases. A special treatment where the samples were inserted into polished sections of the polymer which the surface was polished, and a thin carbon foil was applied using a Balzers CED030 metal coater.

Pycnometry analysis were carried out to measure the rate of porosity within the first- and second-generation composites (2 or 3 rectangles of composites for each measure, Micrometrics AccuPyc 1330, Verneuil en Halatte, France).

#### 2.4.4. Static and Dynamic Mechanical Characterization of Composite Materials

##### Mechanical Tensile Test

Composites were cut with a circular saw with dimensions of 125 mm × 25 mm for first- and second-generation composites. Aluminium stubs measuring 25 mm × 25 mm × 2 mm were glued onto composite specimens previously sanded. Five tests were performed for each first and second-generation composite.

To measure the mechanical capacities in terms of rigidity, i.e., the elastic behaviour of the material, and the breaking strength, tensile tests were carried out according to the ISO 527-4 standard on MTS Criterion 50 (Créteil, France) equipped with a 100 KN load cell. All the tests were performed at a speed of 10 mm/min until the test piece broke. The variation in elongation was measured by a laser extensometer. The higher the elongation, the more ductile the material.

##### Dynamic Mechanical Analysis

Dynamic Mechanical Analysis was used to determine the thermomechanical properties and damping of a material according to the standard ASTM D5026. Analyses were performed on first- and second-generation composite samples (dimensions 10 mm × 50 mm × 1.68 mm) in triplicate from room temperature up to 180 °C at a constant frequency of 1 Hz and a dynamic displacement of 10^−6^ µm (DMA 50N O1-db, Limonest, France).

Linearity tests and a frequency sweep between 0.1 and 100 Hz at room temperature were carried out to verify the viscoelastic properties of the material and select the right displacement.

## 3. Results and Discussion

### 3.1. Dissolution Kinetics

The aim of this section is to compare the effectiveness of different dissolution protocols in terms of dissolution percentage (%*D*) and residual resin rate (%*RR*) on basalt fabrics. The effect of an ultrasonic stirring and the influence of the composite/acetone ratio will be discussed. Ultrasonic stirring is then compared to the application of mechanical stirring.

#### 3.1.1. Influence of an Ultrasonic Stirring Application

An ultrasonic stirring system was applied at different key steps of the dissolution: at the beginning, and at the end of the dissolution, for a “composite:acetone” ratio of 1:4. The efficiency of the dissolution is reported in Figure 3. In each protocol considered below, 3 min of ultrasound were applied. Dissolution profiles are compared with a sample only soaked into acetone (Figure 4).

When no stirring or heating is applied, the induction time is approximately 72 h and the associated %*D* is 81.3% ± 3.7% (Figure 4). Furthermore, resin clogging can be observed at dissolution times of more than 24 h, due to plasticization of the polymer with the solvent [23].

According to Figure 3, for times less than 24 h, a significant difference is observed according to the fact that ultrasounds are applied at the beginning or at the end of dissolution. Applying ultrasounds at the end of dissolution is more efficient than ultrasound at the beginning of dissolution for a dissolution time of 7 h (78% versus 62% of dissolution, respectively). This trend is reversed from 16 h onwards. Furthermore, the average %*D* in the case of ultrasound at the beginning of dissolution are higher than those in the case without ultrasound whatever the time of dissolution, compared to the case with ultrasound at the end of dissolution. This proves the effectiveness of ultrasound in general on resin dissolution kinetics. The standard deviation corresponding to each average %*D* value depends on the capability of the resin to clog on the container. This phenomenon tends to decrease while the solvent volume increases or the dissolution time increases or a mechanical stirring system is applied.

Thus, ultrasound at the beginning of dissolution has a real efficiency for a dissolution time of 16 h, whereas ultrasound at the end of dissolution shows an efficiency on the dissolution of the resin for a dissolution time of 7 h.

The interest of ultrasound at the beginning of the dissolution is that it allows to limit the damage of the basalt fabrics (Figure 5a). Only a few fibers on the fabric edges were disentangled. This is also the case for simple soaking of the samples. On the contrary, Figure 5b shows tufted fibers after dissolution with ultrasound applied at the end of the dissolution. The size of the samples must be considered. It should be noted that the composite surface influences the fabric damage. Thus, if larger samples are considered, it is possible to recover some undamaged fabrics by removing the damaged fabric edges. Figure 5c shows less-damaged fibers; nevertheless, the textile layers stick together because of higher %*RR*. Thus, these recovered fibers are less interesting in terms of reprocess ability after recycling.

#### 3.1.2. Influence of the “Composite:Acetone” Ratio

The influence of the “composite:solvent” ratio was studied in the case of the application of ultrasound at the end of dissolution for 3 min. Thus, the evolution of the %*D* as a function of the dissolution time is plotted in each case and allows to understand the dissolution kinetics of the resin, as shown in Figure 6. Figure 7 allows to observe the relative visual aspect of dissolved resin after solvent evaporation. The results show that the average %*D* increase with the dissolution time whatever the “composite:solvent” ratio even by considering the high standard deviation. Moreover, the higher the solvent volume, the higher the average %*D* value. As explained by Miller-Chou and Koenig [23], the dissolution of a polymer into a solvent are results of two main transport processes: solvent diffusion and chain disentanglement. In contact with a thermodynamically compatible solvent (PMMA with acetone in our case), the solvent diffuses into the polymer. Firstly, a gel-like swollen layer is formed (Figure 7a). Then, after a certain induction time, the polymer is dissolved (Figure 7c).

A comparison of the three ratios: 1:4, 1:10, and 1:40 makes possible to determine the most interesting one (Figure 6), in the case of ultrasound at the end of dissolution. After 7 h of dissolution, there was no significant difference between the average dissolution rate for the ratios 1:4 and 1:40, which allows to reach a satisfactory %*D* over 60%. From a duration of 16 h onwards, the profiles of dissolution become more distinct. When the 1:40 ratio already achieves 100% dissolution, the 1:10 and 1:4 ratios only allow to reach 80% and circa 60% of dissolution, respectively. At times greater than 48 h, the average %*D* value converges towards over 80%, even close to 100% for 1:40 and 1:10 ratios. Some %*D* values are higher than 100% and this is attributed to a slightly accumulation of resin on the glassware after several dissolutions.

A ratio of 1:4 and to a lesser extent a ratio of 1:10 are interesting dissolution conditions for economic and environmental concerns for an upscaling production because of the reduction of solvent used. However, the kinetic of dissolution is then clearly lower, especially in the case of the 1:4 ratio. For this latter, 48 h are required to achieve an average %*D* over 80%. Thus, the 1:10 ratio appears to be a good compromise in terms of dissolution kinetic and volume used with 80% of dissolution reached after 16 h.

#### 3.1.3. Influence of the Mechanical Stirring

The average dissolution profile in Figure 8a shows an average %*D* close to 80% after 3 h of dissolution and a complete dissolution for a dissolution time of 7 h. Slightly higher values than 100% are, as previously, attributed to the fact that small amounts of resin are accumulated on the glassware during the dissolution process, despite careful washing. Then %*D* values decrease for dissolution times above 7 h. The optimal dissolution time required for total dissolution of the resin in this protocol is about 7 h. After recycling, no damage of the fabrics (Figure 8b) is revealed.

#### 3.1.4. Summary of Dissolution Experiments

A simple dissolution in acetone of Elium^®^/basalt fabrics requires a too long dissolution time (72 h) in the case of ratio of 1:4. To reduce the dissolution time, it would be necessary to increase largely the volume of solvent, which is not advantageous from an environmental point of view. The device using ultrasound at the beginning of the dissolution at a ratio of 1:4 makes it possible to combine dissolution efficiency (82.8% ± 6.7%), reasonable dissolution time (16 h), and low damaged fabrics. Later in this paper, the protocol using ultrasounds is adapted to reduce the dissolution time: the dissolution lasts 7 h at a ratio of 1:40 of acetone using 3 min of ultrasounds at the beginning. The device comprising mechanical agitation alone is even more efficient than the previous device: a %*D* of 105.5% ± 13.3% after 7 h of dissolution with a ratio of 1:40 in acetone is reached with no fabric damage. This device requires a larger quantity of solvent but the agitation with the paddle allows a good diffusion of the solvent between the fabric layers. It would also be interesting to compare the energy consumption of a protocol using 3 min of ultrasonic bath and one using 7 h of continuous mechanical agitation. In addition, the acetone/dissolved resin mixture can be processed by evaporation and therefore the acetone can be easily recycled. Thus, a complete recycling cycle of the system can be set up to limit solvent consumption.

### 3.2. Characterization of the Recovered Basalt Fibers

Two methods are used to characterize the residual resin on the surface of the recycled fibers: loss on ignition in quantitative terms and SEM observations in qualitative terms.

#### 3.2.1. SEM Observations of Recovered Basalt Fabrics

SEM imaging is used to confirm the presence of resin on the surface of the fibers after recycling. To understand the influence of the dissolution process on the %*RR*, a comparison is made between virgin basalt fibers and basalt fibers after resin dissolution in presence of acetone (1:4 ratio) for 24 h (Figure 9). A residual resin layer can be easily observed on the recycled fibers, in comparison with virgin fibers.

Figure 10 shows SEM observations of basalt fabrics after dissolution in different conditions. The red circles indicate the localization of residual resin corresponding to the darker area. Figure 10a shows that a very small amount of residual resin remains in the center of a mesh and between the fibers. Acetone seems to have better dissolved the resin on the surface of the fabric layers than between the fibers of the same fabric. This is confirmed by a manual pliability test and the ability to separate fiber layers just after the dissolution. Layers are harder to bend when there is a film of residual resin. The solvent diffusion is in fact harder in the bulk of the composite sample, so that a higher quantity of resin remains in the bulk. However, mechanical stirring tends to limit that phenomenon. On the rest of the recycled fabric, no trace of residual resin was reported.

In the case of recycling with ultrasound at the beginning of dissolution (Figure 10b), clusters of resin are present in the center of the fabric mesh and in slightly greater quantities than in the case of recycling with ultrasounds at the end of dissolution (Figure 10c). In both cases, acetone dissolves the resin better at the crossings between the fibers of the same fabric.

#### 3.2.2. Residual Resin Rate on Basalt Fabrics

This loss on ignition test is performed on basalt fabrics after resin dissolution in different conditions but for a common ratio of 1:4.

According to Figure 11, in the three conditions, the average %*RR* is stable around 2%. However, it remains higher in the case of recycling with ultrasound at the beginning of dissolution, especially at 7 h where the %*RR* is very high (11%) and where the standard deviation is also very high. From 16 h onwards, there is no great difference in the evolution of the residual resin rate according to the dissolution conditions.

From these results, no real effectiveness of ultrasound on the residual resin rate can be discerned. Moreover, in the case of recycling without ultrasound, the residual resin content is always lower than in the case of recycling with ultrasound.

In general, the residual resin amount onto fibers decreases with the dissolution time, which is consistent with the average dissolution profiles (Figure 3). Ultrasound does not significantly reduce the residual resin content on fibers. However, the use of ultrasound at the beginning of the dissolution process seems to increase the average %*D,* regardless of the dissolution time compared to a device without ultrasound. Ultrasound seems to improve the recovery of the resin in the medium without necessarily decreasing the rate of residual resin on the surface of the fibers.

In the case of recycling with mechanical stirring, a very small amount of resin is on the fibers, even at 3 h of recycling (Figure 12), which is confirmed by SEM analysis.

### 3.3. Characterizations of Recovered Resin

#### 3.3.1. ATR FTIR

Infrared analysis allows to understand the influence of acetone dissolution on the chemical composition of the resin. A simple dissolution was performed for 24 h at a “composite:acetone” ratio of 1:4 (Figure 13). The reference peak is the peak at 1720 cm^−1^, which is associated with the C=O bonds from the methacrylate functional group.

An identification of the peaks allows to find common chemical functions between recycled and virgin Elium^®^. The peak at 1140 cm^−1^ is associated with the C–O–C bond, at 1450 cm^−1^ with the C–H bond (bending), and especially the four distinct peaks of CH3 and CH2 bonds (C–H stretching) between 2800 and 3000 cm^−1^ into a consolidated envelope curve.

It can be observed that the three peaks at 1361 cm^−1^, 911 cm^−1^, and 715 cm^−1^ have disappeared in the spectrum of the recycled Elium^®^. The first peak is attributed to C–H Csp3 bonds and the second to C–H Csp2 bonds. Otherwise, the other peaks in the two spectra merge. In addition, small chains, monomer residues, and some additives such as citral can probably exfoliate and migrate out of the polymer. In conclusion, the chemical composition of Elium^®^ recovered after acetone dissolution is close to that of virgin Elium^®^.

#### 3.3.2. SEC

The dissolution of resin samples in acetone coupled with mechanical stirring or ultrasounds may have an influence on the chemical structure of the polymer. Size exclusion chromatography allows to understand whether a polymer degradation can occur. Thus, the evolution of Mn, Mw, and also the polydispersity (Table 2) and molecular weight distribution (Figure 14) helps to get an overview of degradation mechanisms during dissolution.

In both recycling cases, PD increases. The molecular weight distribution curves do not show the formation of smaller molecules but rather larger ones in the case of recycling with ultrasounds. Thus, in that case, recombination or intermolecular reactions may take place. After recycling, the molecular weight distribution remains unimodal. In the case of recycling with mechanical stirring, the size of molecules is similar to that of virgin Elium^®^.

In the case of mechanical stirring alone, Mw remains constant and Mn decreases (−28%). Thus, this stirring system does not imply any degradation of the polymer chains but allows a faster disentanglement of the chains during solvation.

In the case of recycling with ultrasound at the beginning of dissolution, Mw (−21%) and Mn (−11%) decrease. Thus, the polymer could degrade and involve the creation of smaller molecules. According to [23], only long chains can be affected by ultrasound. Thus, the heterogeneity in terms of molecular weights decreases. Indeed, the polydispersity PD remains close to that of virgin Elium^®^. However, as ultrasound is only applied for 3 min at the beginning of dissolution, it is not sufficient to really create a degradation. Recombination of molecules can also be at the origin of the formation of larger molecules.

### 3.4. Mechanical Properties of Second-Generation Composites

#### 3.4.1. Static Tensile Tests

Mechanical results of uniaxial tensile tests performed on Elium^®^/basalt composites are reported in Table 3. The pore volume of the composites was also measured and are presented in Table 3.

Second generation composites after a dissolution process using mechanical stirring have a 22% higher modulus (24.4 ± 4.7 GPa against 19.9 ± 2.7 GPa in the initial state) and a 15% higher stress (586.8 ± 22.1 MPa against 508 ± 34.8 MPa). After a dissolution process using ultrasound, second generation composites have tensile properties comparable to first generation composites with a modulus (−3%) of 20.6 ± 2.4 GPa against 19.9 ± 2.7 GPa and a stress of 474.3 ± 24.5 MPa against 508.0 ± 34.8 MPa (−6%). The lower fabric degradation after dissolution process using mechanical stirring (Figure 6) compared to those using ultrasound (Figure 5) may explain the better performance in tensile tests.

However, for dissolution conditions, the porosity is lower than for first generation composites, −51% for mechanical stirring despite an equal fiber weight ratio ≅ 71.4%. Nevertheless, composites from ultrasounds have less than −66% of porosity with a fiber weight fraction much lower (67.1 ± 0.8%). In addition, despite a higher porosity rate (4.2% by mechanical stirring vs. 2.9% by ultrasounds), the composite recycled by mechanical stirring has superior tensile performance (+15% in modulus and +19% in stress), compared to the composite recycled by ultrasounds.

In addition, the SEM images (Figure 15a–c) show no difference in interfacial adhesion between the two types of recycling—good impregnation and interfacial adhesion, which contribute to the good mechanical properties of second-generation composites.

#### 3.4.2. Dynamic Mechanical Analysis

The tests were carried out on first- and second-generation composites (Figure 16). For the latter, the dissolution was performed using ultrasound for 3 min in the beginning for 16 h (ratio 1:4) and mechanical stirring for 7 h (ratio 1:40). Table 4 is a summary of the data measured from the DMA tests, where E’160 and E’40 are the rubber and vitreous conservation modules, respectively.

The first generation composite shows a main relaxation peak related to glass transition temperature located at about 139 °C with an intensity of 0.32 At the same time, the modulus drops from 19.87 ± 1.87 GPa to 4.25 ± 0.69 GPa.

In the case of a dissolution using mechanical stirring, second generation composites provide evidence of a shoulder on the main relaxation peak located at 122 °C, i.e., at a lower temperature (151 °C) than the main relaxation. The intensity of both peaks is 0.13 (shoulder) and 0.21 (main peak). In the case of a dissolution using ultrasounds, no shoulder is observed and the main relaxation is located at about 139 °C as for the first-generation composites. Nevertheless, a higher intensity is determined (0.40). As concerns storage modulus, it drops from 18.99 ± 0.27 GPa to 6.30 ± 0.3 GPa when mechanical stirring is used and from 16.25 ± 1.18 GPa to 3.03 ± 0.20 when ultrasounds are used. First, it can be concluded that the storage modulus at 40 °C is not affected by recycling and corresponds to the same order than the first-generation composites.

tan dα Differences may be explained by the presence of residual resin on the fabrics after resin dissolution. This resin may form a thin homogeneous layer when mechanical stirring is used, as it seems to be a more heterogeneous cluster when ultrasounds are used (Figure 10), even if the residual resin rate is higher in the case of ultrasonic recycling in comparison with the mechanical method. Further investigation will be carried out to confirm this assumption.

## 4. Conclusions

The objective of this work was to compare the material recoveries resulting from different chemical recycling methodologies for thermoplastic acrylate-based composites reinforced by basalt fabrics and manufactured by vacuum infusion. The recycling was processed via chemical dissolution with a preselected adapted solvent i.e., acetone. The main goal of the study was to recover undamaged basalt fabrics to reuse them as reinforcements for second generation composites.

A protocol based on an ultrasound technique was compared to another one using mechanical stirring. The results show that the dissolution protocol using a mechanical stirring is more adapted to recover undamaged fabrics with no residual resin on their surface.

Several parameters, such as dissolution duration, dissolution temperature, and solvent/composite ratio were also studied. The results show that in the case of a 1:40 ratio, the optimal dissolution time required for total dissolution of the resin is about 7 h. Moreover, the higher the solvent volume, the higher the dissolution rate.

FTIR and SEC analysis show that there is no degradation of the resin after dissolution by acetone, but recombination phenomena do not modify the chemical composition of the Elium^®^ resin. The stirring system allows disentanglement of long polymer chains, and this has a low impact on the molecular mass distribution.

Finally, second generation composites were elaborated with these recycled fabrics and mechanical and thermomechanical properties were determined and compared to those of first-generation composites. Corresponding second generation composites displayed equivalent mechanical properties than first generation ones.

Heating was avoided in this study for lower energy consumption and lower safety risks, but it should be interesting to evaluate the effect of temperature. As the protocol with mechanical agitation requires a larger amount of solvent (50 mL of acetone for 1 g of composite), whereas the protocol with ultrasound can be limited to 5 mL of solvent for 1 g of composite, an up-scale study should be performed. Greener solvent should be considered in further investigations.

## Figures and Tables

**Figure 1 polymers-14-01083-f001:**
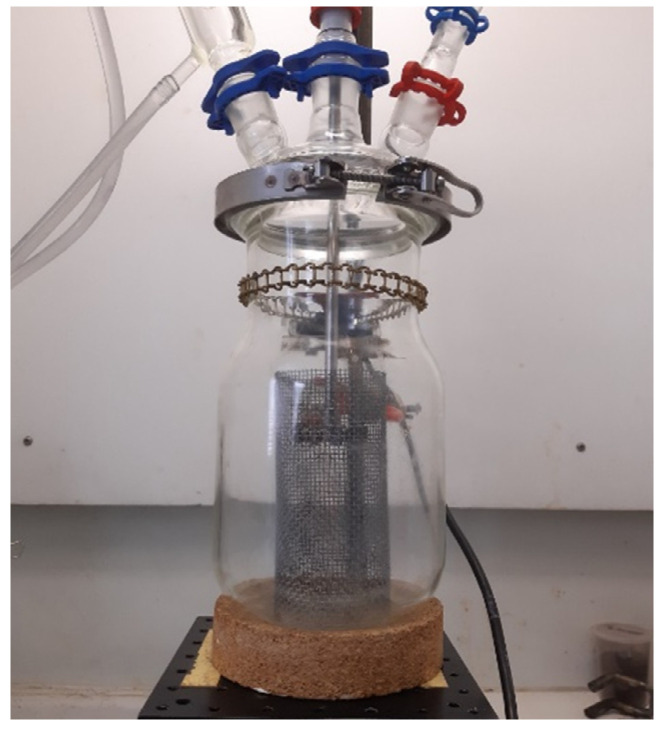
Experimental device used for dissolution experiments using mechanical stirring.

**Figure 2 polymers-14-01083-f002:**
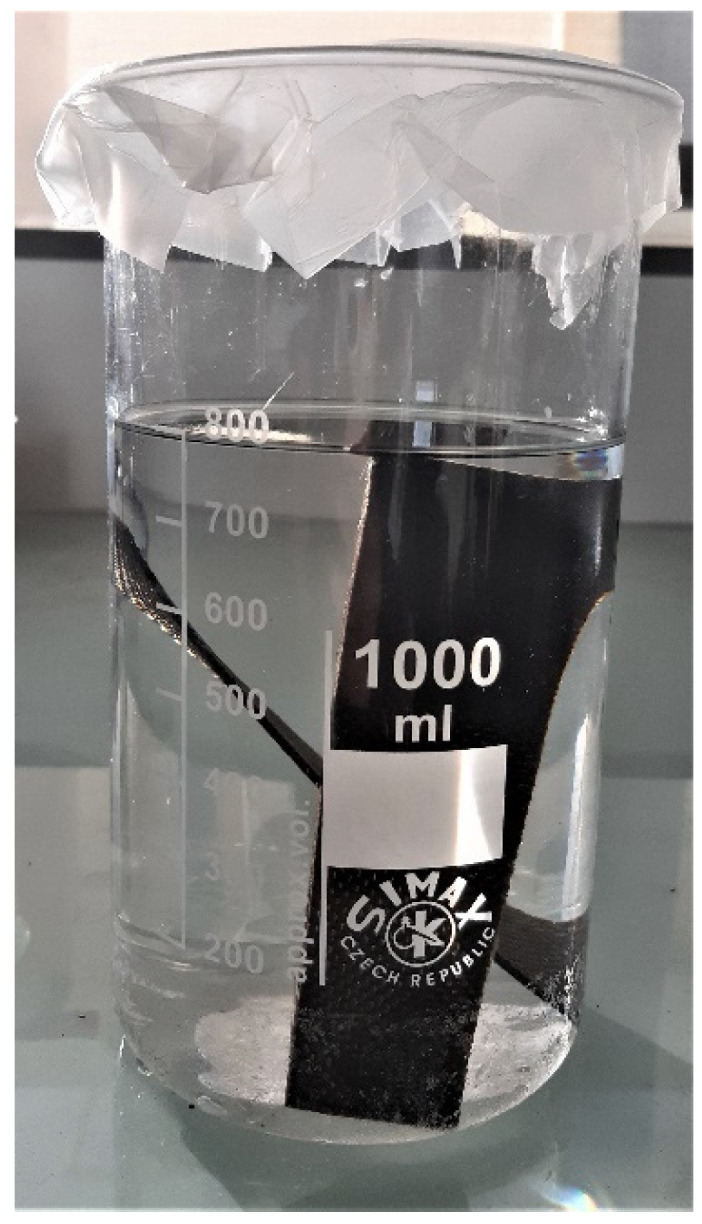
Experimental device used for dissolution experiments using ultrasounds.

**Figure 3 polymers-14-01083-f003:**
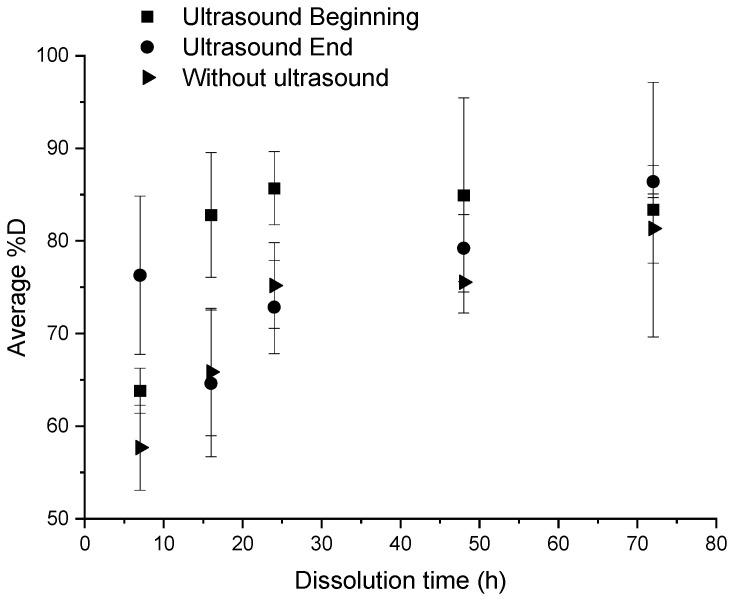
Influence of different ultrasonic conditions on the average %*D* as a function of dissolution time for a “composite:acetone” ratio of 1:4.

**Figure 4 polymers-14-01083-f004:**
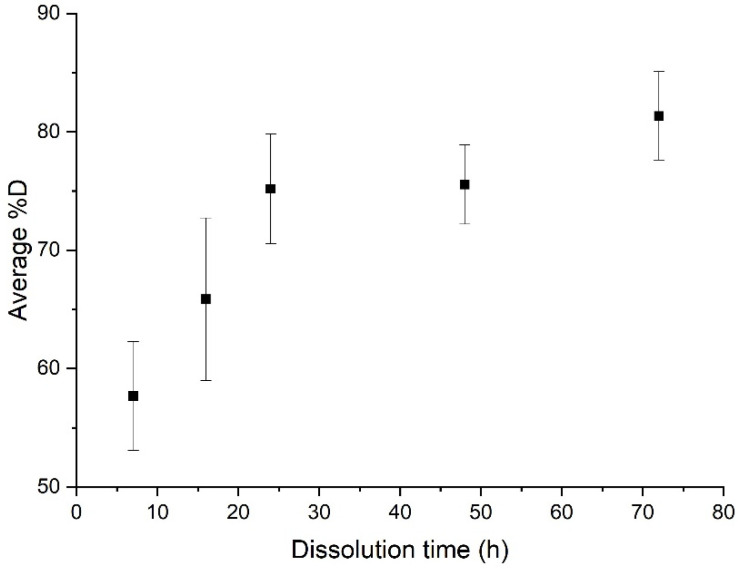
Evolution of the average %*D* as a function of dissolution time for a simple dissolution of a “composite:acetone” ratio of 1:4.

**Figure 5 polymers-14-01083-f005:**
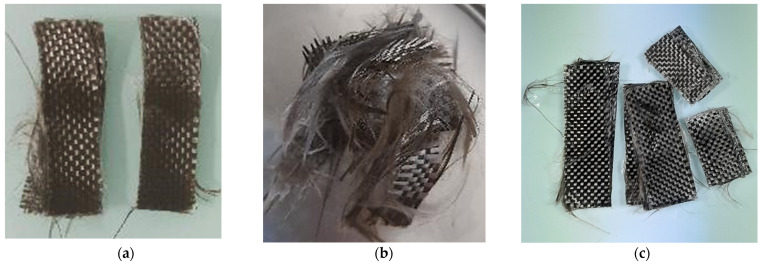
Basalt fibers recovered (**a**) after recycling 16 h with ultrasound at the beginning of the dissolution, (**b**) after recycling 7 h with ultrasounds at the end of the dissolution and (**c**) after recycling without ultrasounds. A 1:4 ratio is considered for each photo.

**Figure 6 polymers-14-01083-f006:**
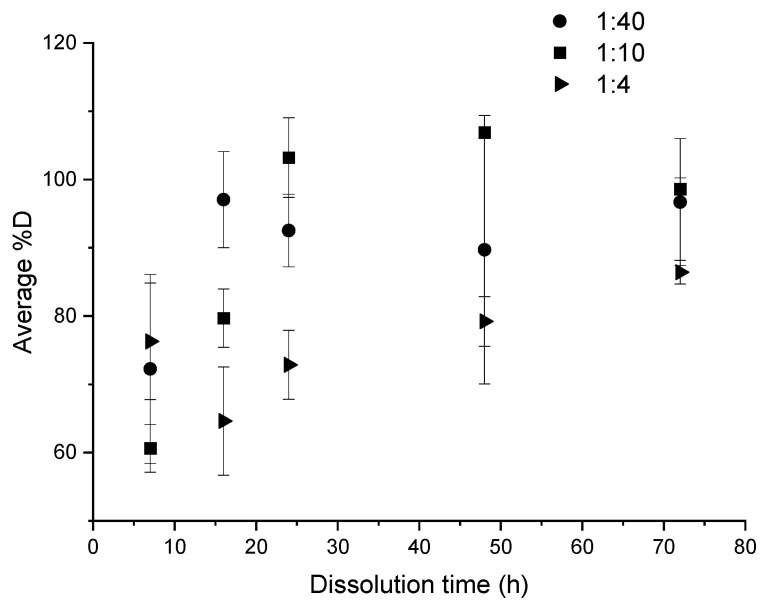
Influence of “composite:acetone” ratio on the average %*D* as a function of dissolution time for ultrasound at the end of dissolution.

**Figure 7 polymers-14-01083-f007:**
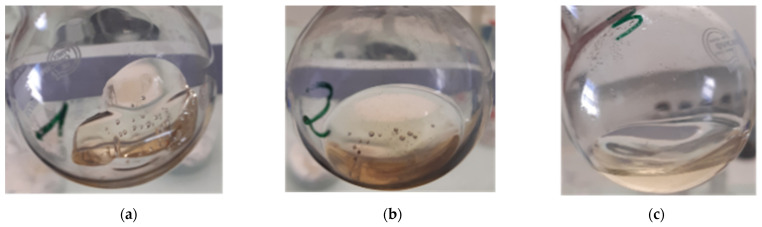
Dissolution of Elium^®^ resin during 24 h with acetone at different “resin:solvent” ratios: (**a**) 1:1, (**b**) 1:2 and (**c**) 1:4.

**Figure 8 polymers-14-01083-f008:**
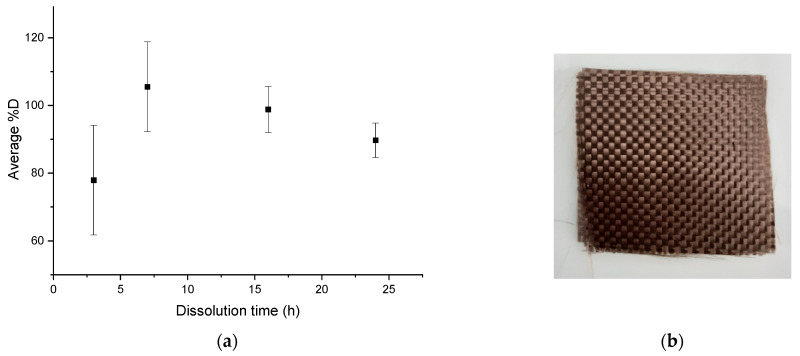
(**a**) Evolution of the average %*D* as a function of dissolution time for mechanical stirring for a ratio of 1:40, (**b**) Fabrics obtained after the complete dissolution.

**Figure 9 polymers-14-01083-f009:**
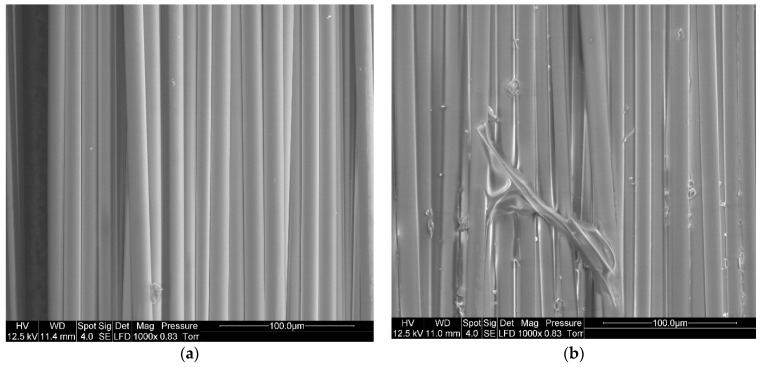
SEM image of (**a**) Virgin basalt fibers, (**b**) Recycled basalt fibers dissolved in acetone for 24 h at a solvent ratio of 1:4.

**Figure 10 polymers-14-01083-f010:**
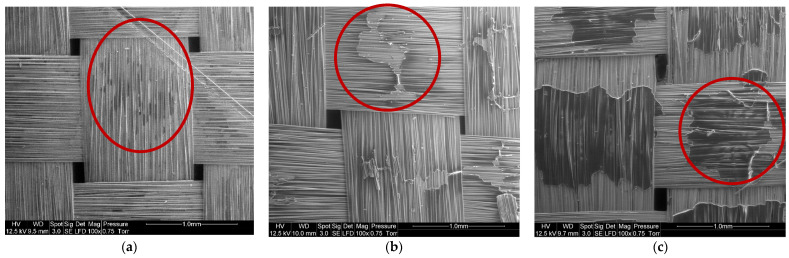
SEM image of (**a**) Basalt fibers recovered by mechanical stirring (1:40 ratio) during 7 h, (**b**) Basalt fibers recovered with ultrasound at the beginning of the dissolution (1:4 ratio) during 16 h and (**c**) Basalt fibers recovered with ultrasound at the end of the dissolution (1:4 ratio) during 7 h.

**Figure 11 polymers-14-01083-f011:**
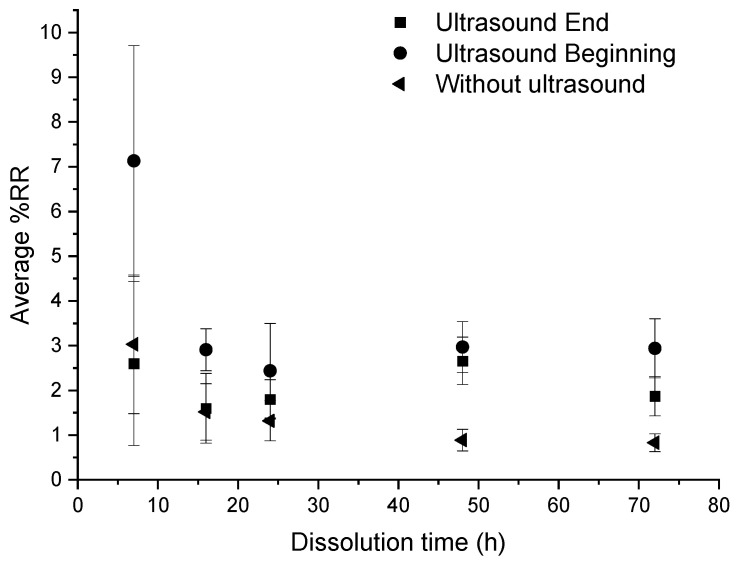
Influence of different ultrasonic conditions on the average %*RR* as a function of dissolution time for composite: acetone ratio of 1:4.

**Figure 12 polymers-14-01083-f012:**
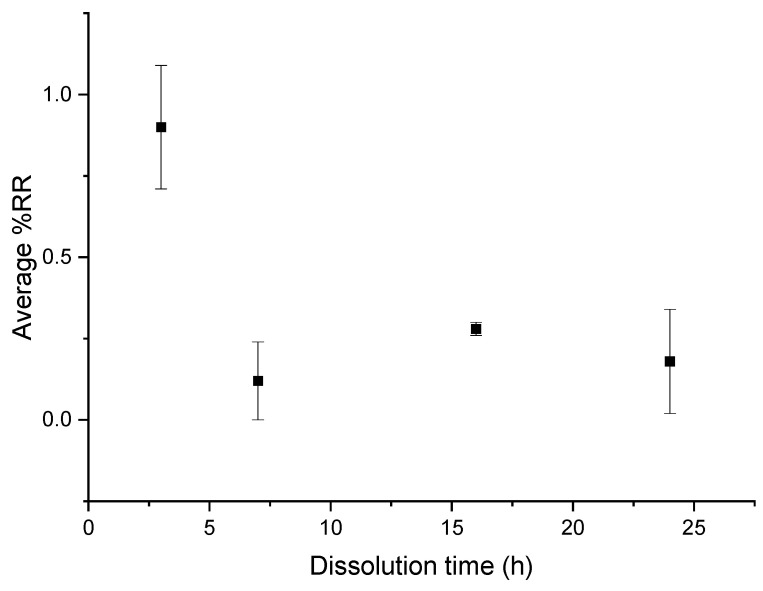
Evolution of the average %*RR* as a function of time for recycling with mechanical stirring and acetone at a ratio of 1:40.

**Figure 13 polymers-14-01083-f013:**
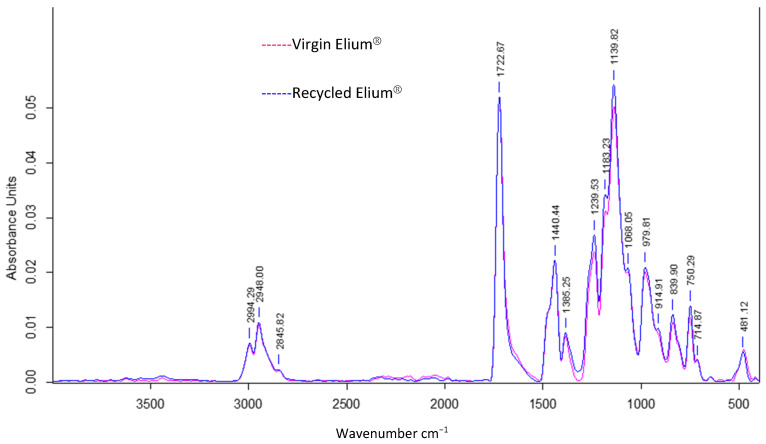
ATR FTIR absorbance spectrum of virgin Elium^®^ and Elium^®^ after 24 h of simple dissolution in acetone 1:4.

**Figure 14 polymers-14-01083-f014:**
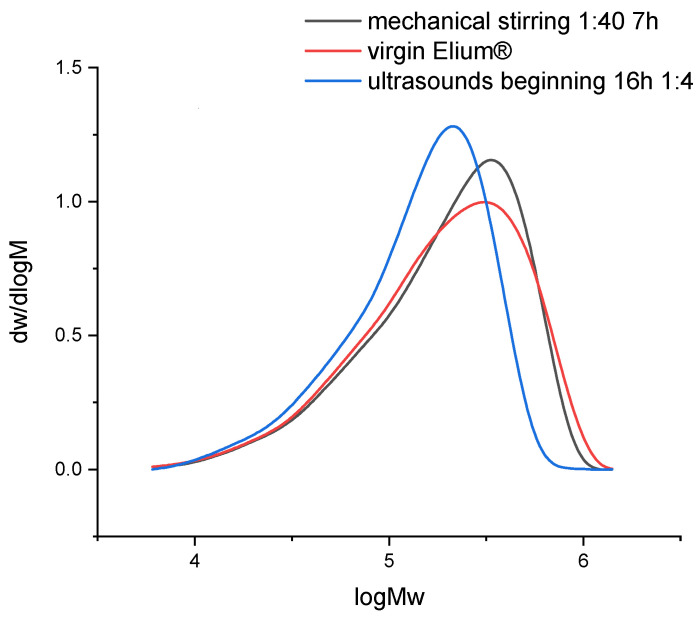
Molecular weights distribution of virgin Elium^®^ and recycled Elium^®^ resin after dissolution.

**Figure 15 polymers-14-01083-f015:**
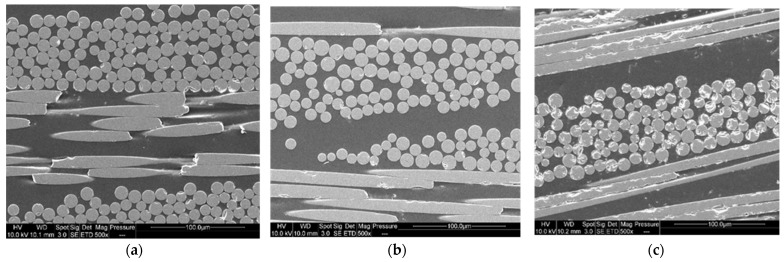
SEM images of (**a**) first generation composite (**b**) second generation composite after dissolution using mechanical stirring and (**c**) second generation composite after dissolution using ultrasound in the beginning.

**Figure 16 polymers-14-01083-f016:**
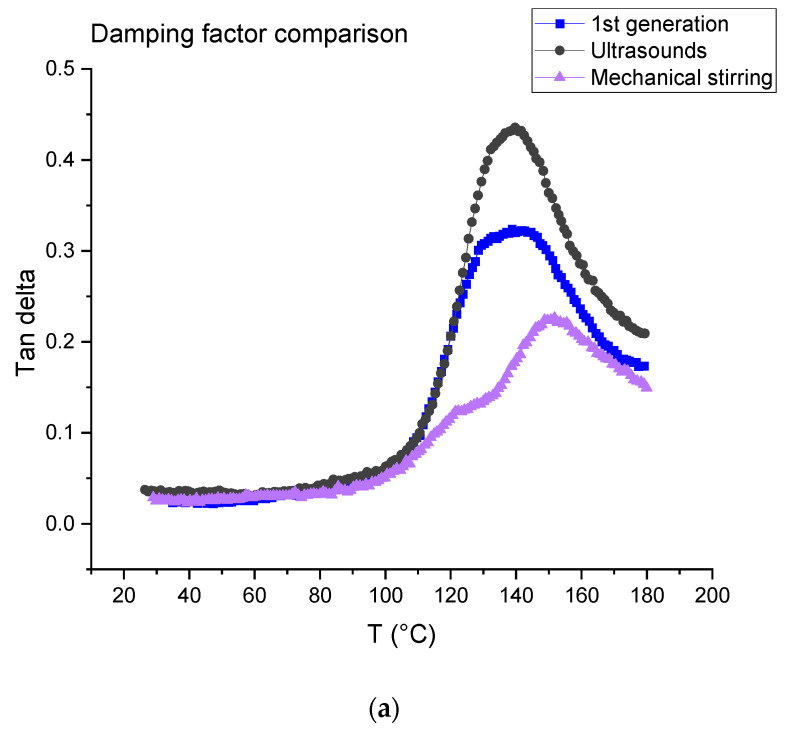
(**a**) Tangent delta; (**b**) conservation modulus of first-generation composites and second-generation composites obtained after dissolution using mechanical stirring and ultrasounds.

**Table 1 polymers-14-01083-t001:** Conditions of dissolution.

Parameters	Conditions
Samples dimension	6 cm × 10 cm or 13.5 cm × 3.5 cm
Container	1 or 2 L reactor
Solvent	Acetone 99.9% purity
“Composite:solvent” ratio	1:4, 1:10, 1:40
Dissolution time	7, 16, 24, 48, 72 h
Temperature	25 °C
Mechanical agitation	Pale notched, 60 rpm
Ultrasounds	3 min applied at the beginning or end of dissolution

**Table 2 polymers-14-01083-t002:** Average molecular weights and polydispersity of recycled and virgin resin after dissolution.

	Mn (g/mol)	Mw (g/mol)	PD
Ultrasounds at the beginning acetone 1:4 16 h	67,790	189,140	2.8
Mechanical stirring acetone 1:40 7 h	61,890	217,440	3.5
Virgin Elium^®^	85,600	213,200	2.5

**Table 3 polymers-14-01083-t003:** Modulus, tensile strength, and porosity of first and second generation Elium^®^/basalt composites recycled using mechanical stirring and ultrasounds at the beginning of the dissolution.

	Modulus (GPa)	Stress (MPa)	Porosity (%)	Fiber Weight Fraction (%)
1st generation	19.9 ± 2.7	508.0 ± 34.8	8.6 ± 5.9	71.4 ± 1.0
2nd generation: mechanical stirring	24.4 ± 4.7	586.8 ± 22.1	4.2 ± 2.7	71.4 ± 0.4
2nd generation: ultrasounds at beginning	20.6 ± 2.4	474.3 ± 24.5	2.9 ± 2.0	67.1 ± 0.8

**Table 4 polymers-14-01083-t004:** DMA results of first generation composites and those recycled by mechanical stirring and ultrasound.

	1st Generation	2nd Generation Mechanical Stirring	2nd Generation: Ultrasounds
E’40 (GPa)	19.87 ± 1.87	18.99 ± 0.27	16.25 ± 1.18
E’160 (GPa)	4.25 ± 0.69	6.30 ± 0.35	3.03 ± 0.20
tan d1	/	0.13 ± 0.01	/
T1 (°C)	/	122.30 ± 1.45	/
tan dα	0.32 ± 0.01	0.21 ± 0.02	0.40 ± 0.03
Tα (°C)	139.27 ± 2.12	151.11 ± 1.16	138.81 ± 4.97

## Data Availability

Data cannot be shared, as it is a part of an ongoing study.

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
