# Peer review of "Chemical Recycling of Vacuum-Infused Thermoplastic Acrylate-Based Composites Reinforced by Basalt Fabrics"

_polymers, 2022, doi:10.3390/polym14061083_

Round 1

Reviewer 1 Report

Please find the enclosed file

Author Response

Comments to Author

N: Dear Authors, I have keenly analyzed your article, and I have observed there is a lot of information in the article, which needs to be organized smoothly. There are a huge number of figures but the reader loose track while go through the data. If you are given the chance of revision. Please arrange it in a technical way, The scales of the figures are slightly blur in some figures, you can paste the graph in its original form to avoid visibility issue. Please find the comments below to improve the article.

Answer: We would like to thank you for having reviewed our article. We took note of the various comments you made about it and we took care to make the changes needed.

Minor correction

Abstract

Comment 1: Line 11 “The object of this work is to compare the material recoveries resulting from different chemical recycling” It is better to revise as, The objective of this work is to compare the material recovered from…

Answer: Thanks for indicating this, we have corrected it.

Comment 2.

Introduction:

There is no need of headings in the introduction.

Answer: We think the headings help to be clearer and to structure the approach in this article. So we decided to keep them.

Experimental:

Major

Comment 1

It seems the authors have synthesized the composites in the section “Composites processing”? Were the samples freshly prepared for recycling purpose? If yes? Is it appropriate to use the freshly prepared sample for recycling? Will there be any difference between the freshly prepared recycled samples and recycling of actually used sample?

Answer: Indeed the aim of this article is to recycle fresh materials and to study the influence of various dissolution parameters on composites at the beginning of their lifecycle. It would also be interesting to study the recycling of aged composites. It takes time to recover aged composites and it is the subject of a current thesis. That would be the subject of another article.

Major 

Comment 2

Authors should provide the reference in the experimental section.

Answer: In the experimental section, we did not put any reference because the tests are standardized and so we chose to quote the standards (NF EN ISO 1172 for loss ignition tests, ASTM D5026 for Dynamic Mechanical Analysis) followed with the test conditions adapted to our materials.

Comment 3

Is there any ASTM which have been used for recycling? Or any other standards?

Answer: To carry out the recycling tests, we were inspired by the literature and existing works, but there are no standards to refer to.

Comment 4

How the Dissolution devices have been developed? How the parameters have been selected?

Answer: First of all, we have listed the solvents in which the Elium 150 resin could be soluble and the least dangerous possible (environment, toxicity). Then we were inspired by what has already been done in the literature to carry out preliminary tests and the means available in the laboratory.  From these results we selected a solvent (acetone) and the most suitable experimental conditions (dissolution time, temperature, stirring system).

Comment 5

Figure 2 explanation is missing in text, apparently it seems a beaker, it needs explanation which kind of device it is?

Answer: Yes, this was done in a beaker, and was added in the body of the text

Minor

Comment 1

Author should use the past tense while reporting the experimental section in particular such as in mechanical testing.

Answer: Yes, this has been changed throughout the text

Comment 2

Line 263: Table title should be above table

Answer: All table titles have been moved to the top of the tables.

Comment 3

All the equations need references

Answer: The equations used are derived from manipulations that have been performed according to cited standards, but there is no specific reference associated. However we have numbered the equations for a better reading.

Results and Discussion

Major

“Dissolution kinetics” & “Influence of an ultrasonic stirring application” This should be included in Experimental section.

Answer: We chose to put the following section: “Dissolution kinetics and influence of an ultrasonic stirring application” in Results and discussion because there are fully considered as experimental results.

Figure 3 and 4 needs explanation. Author should use different symbols for better understanding, apparently all square are similar, making it hard to differentiate.

Answer: Figure 3 and 4 have been further explained Line 405 and 408.

Symbols have also been changed. Also the Standard error appears quite large in some values? Why it is so?

Answer: At very low solvent volumes, Elium resin tends to be more viscous so that it clogs on the glass of the container. Thus, the standard deviation for some dissolutions is higher.

We attempt to reduce SE of %D however it should have been needed to add some %D results to confirm this trend. Figure 5 c, The results show better Basalt fibers recovered without ultrasound, author should comment and provide explanation.

Answer: Basalt fibers without ultrasounds have less damages nevertheless they have a greater %RR and fiber layers are sticking together so that’s less interesting. (Line 430-432)

Line 453 This conclusion section should not be here, it is confusing for the reader.

Answer : The aim of this conclusion is to set the context of further experimental results. It sums up and justifies the choices that have been made for further results regarding mechanical properties.

To clarify we changed the heading: “Summary of dissolution experiments”.

Line 472; “Characterization of the recovered basalt fibers”

This part should be in the experimental section.

Answer: We decided to present both results and discussion in the same time instead of separating the results on the one hand and to put the discussion after.

Figure 9 & 10. The scale on SEM images in unreadable.

Answer: Thanks for this remark, this has been changed.

Fig 11: The SE should be rechecked.

Answer: We should add some %RR results to confirm the SE value (in general we did 3 values for each %RR).

Line 535, This characterization part should be in the experimental section

Answer: We decided to present both results and discussion at the same time instead of separating the results on the one hand and to put the discussion after.

All the titles of the tables should be placed above table

Answer: Thanks for the remark, all table titles have been moved to the top of the tables.

Author should place the numbering of all the headings of results and discussion

Answer: Thanks for this comment and we think that all the headings have now been numbered.

Figure 14, scale is not clear

Answer: Thanks for your remark, we have modified this.

Figure 16 a No need to write tan delta without unit, only tan delts is enough.

Answer: The figure has been modified

References

The references should be according to Journal’s style.

Answer: Changes were made to references that did not follow the style of the journal.

Reviewer 2 Report

This paper describes a comparison of the material recoveries resulting from different recycling methods for thermoplastic acrylate-based composites reinforced by basalt fabrics. The article is interesting and well-structured. The cited references are current, but there is a small mistake with the double numeration of references. The figures like 1,2,5,7 are of poor quality; in the reviewer's opinion, they should be improved. Figures 9 and 10 should include information in the caption that these are SEM photos.

Author Response

This paper describes a comparison of the material recoveries resulting from different recycling methods for thermoplastic acrylate-based composites reinforced by basalt fabrics. The article is interesting and well-structured.

Answer: Thank you for the comment, we appreciate the interest you have against our article.

The cited references are current, but there is a small mistake with the double numeration of references.

Answer: We didn’t find the error you mentioned, sorry. Could you please indicate the reference concerned by the double numeration?

The figures like 1,2,5,7 are of poor quality; in the reviewer's opinion, they should be improved.

Answer: Thanks for your remark, we have modified this.

Figures 9 and 10 should include information in the caption that these are SEM photos.

Answer: The titles of figures 9 and 10 have been modified with the addition that the images are SEM photos.

Reviewer 3 Report

The paper under review is devoted to considering a possibility and basic technology for Chemical recycling thermoplastic acrylate based composites reinforced by basalt fabrics  

I suppose that the aim of the work is quite interesting and the proposed technology can be really explored/.

The paper contains all necessary component of a research work carried out quite accurately with marking possible confidence limits, and the obtained results can be used in practice.

I believe that this manuscript can be accepted for publication as it is presented.

Author Response

The paper under review is devoted to considering a possibility and basic technology for Chemical recycling thermoplastic acrylate based composites reinforced by basalt fabrics  

I suppose that the aim of the work is quite interesting and the proposed technology can be really explored/.

The paper contains all necessary component of a research work carried out quite accurately with marking possible confidence limits, and the obtained results can be used in practice. I believe that this manuscript can be accepted for publication as it is presented. 

Answer: Thank you for this comment which gratifies our research work.